# Comparisons of Videolaryngoscopes for Intubation Undergoing General Anesthesia: Systematic Review and Network Meta-Analysis of Randomized Controlled Trials

**DOI:** 10.3390/jpm12030363

**Published:** 2022-02-26

**Authors:** Juncheol Lee, Youngsuk Cho, Wonhee Kim, Kyu-Sun Choi, Bo-Hyoung Jang, Hyungoo Shin, Chiwon Ahn, Jae Guk Kim, Min Kyun Na, Tae Ho Lim, Dong Won Kim

**Affiliations:** 1Department of Emergency Medicine, Hanyang University College of Medicine, Seoul 04763, Korea; doldoly@hanyang.ac.kr (J.L.); seodtst@gmail.com (H.S.); erthim@gmail.com (T.H.L.); 2Department of Emergency Medicine, Hallym University, Kangdong Sacred Heart Hospital, Seoul 05355, Korea; faith2love@gmail.com; 3Department of Biomedical Engineering, Hanyang University College of Medicine, Seoul 04763, Korea; 4Department of Emergency Medicine, Hallym University, Chuncheon 24253, Korea; gallion00@gmail.com; 5Department of Neurosurgery, Hanyang University College of Medicine, Seoul 04763, Korea; vertex-09@hanmail.net (K.-S.C.); mavmav@hanmail.net (M.K.N.); 6Department of Preventive Medicine, College of Korean Medicine, Kyung Hee University, Seoul 02447, Korea; bhjang@khu.ac.kr; 7Department of Emergency Medicine, College of Medicine, Chung-Ang University, Seoul 06974, Korea; cahn@cau.ac.kr; 8Department of Anesthesiology and Pain Medicine, Hanyang University College of Medicine, Seoul 04763, Korea; dongwkim@hanyang.ac.kr

**Keywords:** laryngoscopes, anesthesia, intubation, systematic review, meta-analysis

## Abstract

Background: The efficacy and safety of videolaryngoscopes (VLs) for tracheal intubation is still conflicting and changeable according to airway circumstances. This study aimed to compare the efficacy and safety of several VLs in patients undergoing general anesthesia. Methods: Medline, EMBASE, and the Cochrane Library were searched until 13 January 2020. The following VLs were evaluated compared to the Macintosh laryngoscope (MCL) by network meta-analysis for randomized controlled trials (RCTs): Airtraq, Airwayscope, C-MAC, C-MAC D-blade (CMD), GlideScope, King Vision, and McGrath. Outcome measures were the success and time (speed) of intubation, glottic view, and sore throat (safety). Results: A total of 9315 patients in 96 RCTs were included. The highest-ranked VLs for first-pass intubation success were CMD (90.6 % in all airway; 92.7% in difficult airway) and King Vision (92% in normal airway). In the rank analysis for secondary outcomes, the following VLs showed the highest efficacy or safety: Airtraq (safety), Airwayscope (speed and view), C-MAC (speed), CMD (safety), and McGrath (view). These VLs, except McGrath, were more effective or safer than MCL in moderate evidence level, whereas there was low certainty of evidence in the intercomparisons of VLs. Conclusions: CMD and King Vision could be relatively successful than MCL and other VLs for tracheal intubation under general anesthesia. The comparisons of intubation success between VLs and MCL showed moderate certainty of evidence level, whereas the intercomparisons of VLs showed low certainty evidence.

## 1. Introduction

The tracheal intubation during general anesthesia can be often unsuccessful. Although the intubation is successful, it can cause several complications. These included respiratory (sore throat, airway trauma), hemodynamic (bradycardia, tachycardia, hypotension) or mechanical complications (mucosal bleeding, dental injury) [1,2]. Difficult intubation occurs in 1.8–5.8% of patients, and failed intubation occurs less frequently but still in 0.13–0.30% of cases [2,3,4].

The Mallampati score, mouth opening, and thyromental distance, as well as body mass index, have all been established as predictors of difficult intubation. When patients are obese, have limited neck movement, have a narrow jaw opening, an enlarged tongue, or have poor tissue mobility, airway difficulties rise. [5,6,7]. Several types of videolaryngoscopes (VLs) have been developed to overcome these difficult airways and make intubation successful by providing better glottic view. VLs are categorized into channeled (without stylet) versus non-channeled (with stylet). Channeled VLs include Airtraq, Airwayscope and King Vision, whereas non-channeled VLs include GlideScope, C-MAC, C-MAC D-blade (CMD), and McGrath. Although it appears that the mechanism of VL is helpful in increasing success rates, several randomized controlled trials (RCTs) have suggested that VLs are surprisingly not more effective than MCL [8,9,10,11,12,13,14]. However, the results of all existing meta-analyses are still conflicting and are limited to the comparison of specific VL and MCL [15,16,17,18,19].

To identify relative superiority among several VLs for patient outcomes, the use of network meta-analysis can be appropriate. It can analyze and rank the efficacy of specific VL comparing with other VL as well as MCL. We aimed to identify the most effective and safest VL for tracheal intubation undergoing surgery considering airway circumstances by performing network meta-analysis. 

## 2. Methods

### 2.1. Literature Search

This systematic review and network meta-analysis of RCTs is based on the Preferred Reporting Items for Systematic Reviews and Meta-analysis (PRISMA). The study protocol was registered with PROSPERO (CRD42019126284).

Two investigators (B.-H.J. and K.-S.C.) constructed the search strategy and other two investigators (Y.C. and K-S.C.) provided formal review. We conducted a search of Medline, EMBASE, and the Cochrane Library from their inception to 30 January 2020. To ensure high sensitivity in our search, we designed search strategies that included pertinent MeSH keywords, common keywords, and their comprehensive combination. The product name of VLs were also included in our search criteria. There were no restrictions on languages, and no filters were applied in this search. Details of the search strategy are described in Appendix A. 

### 2.2. Data Selection

We contrived a question based on population, intervention, comparison, and outcome (PICO). The PICO question was as follows: P—adult patients who required tracheal intubation during general anesthesia for elective surgery; I—VLs having independent video display for indirect vision; C—MCL direct laryngoscope with any VL with video display except intervention; O—the success rate for the first attempt intubation. Patients who underwent emergency surgery while under general anesthesia were not included in the study. The VLs included in this review were Airtraq (Prodol, Vizcaya, Spain), Airwayscope (Nihon Kohden, Tokyo, Japan), C-MAC (Karl Storz, Tuttlingen, Germany), CMD (Karl Storz, Tuttlingen, Germany), GlideScope(Verathon, WA, USA), King Vision(Ambu, Copenhagen, Denmark), and McGrath (Medtronic, Dublin, Ireland) (Figure 1). The following VLs were excluded due to insufficient articles to perform network meta-analysis: Trueview, A.P. Advance, UE scope, UE video intubation stylet. The use of King Vision as a non-channeled VL was excluded. The Optic stylet relying on direct vision through an embedded fiberoptic bundle was excluded.

### 2.3. Data Identification and Extraction

The study data were collected and extracted using a standardized form. Two investigators (W.K. and J.L.) independently screened articles by title, abstract, and full texts according to the prespecified inclusion criteria. A full-text review was subsequently performed for potentially relevant articles. Any discrepancies were resolved by consensus after consulting a third investigator (Y.C.). The inclusion criteria were as follows: (1) adult patients who underwent tracheal intubation by experienced anesthetists during general anesthesia, and (2) RCT studies published in English for VL or MCL. The exclusion criteria were (1) non-relevant intervention, (2) studies that failed to acquire outcomes of interest, (3) non-adult studies, and (4) non-RCTs such as review letters, before and after studies, observational studies, case-control studies, case reports, pre-prints, and conference abstracts.

### 2.4. Outcome Measures

The primary outcome of efficacy was the success rate for first-attempt intubation. The success of intubation was defined by capnography confirmation. Other success outcomes measured by chest rise or visual confirmation using VL were excluded. The secondary outcomes were intubation time, glottic view on the first attempt of intubation, and the incidence of sore throat within 24 h. The intubation time was defined as the time from picking up the laryngoscope to confirmation by capnography. The glottic view was assessed using the Cormack–Lehane grade (CLG, I–IV) or modified CLG. The good-glottic view was also defined as CLG I-II or modified CLG I and IIa [20]. Better efficacy or safety means a higher success rate, shorter intubation time, better glottic view, and lower incidence of sore throat. These four outcomes were evaluated by three categories of airway status (all vs. normal vs. difficult airway). All airways were defined as the normal airway mixed with a difficult airway. Normal airway was additionally defined as airway circumstance that did not predict a difficult airway. Difficult airways were predicted using the following definitions: morbidly obese participants (body mass index > 35 kg/m^2^); patients with immobilized cervical spines; Mallampati classification 4; retrognathia; more than one of the following: Mallampati classification 3, inter-incision distance of 35 mm or less, and a thyromental distance of 65 mm or less [21,22]. 

### 2.5. Quality Assessment

Quality assessment was also independently performed by the reviewers using the risk of bias tool developed by the Cochrane group [23]. Evaluated biases included: (1) random sequence generation; (2) allocation concealment; (3) blinding of participants and personnel; (4) blinding of outcome assessments; (5) incomplete outcome data; (6) selective reporting; and (7) other bias. The methodological quality of the identified studies was assessed independently by W.K. and J.L. Investigators selected the terms “low risk of bias,” “high risk of bias,” or “unclear” to define each study. Any disagreements between the investigators were resolved by a third investigator. 

### 2.6. Reporting Guidelines and Certainty of Evidence

The modified Grades of Recommendation, Assessment, Development and Evaluation (GRADE) tool for network meta-analysis was used to evaluate the quality of evidence [24]. The quality of the results were classified as follows: (1) high quality—further research is very unlikely to change the confidence in the estimated effect; (2) moderate quality—further research is likely to have an important impact on the confidence in the estimated effect and may change the estimate; (3) low quality—further research is very likely to have an important impact on the confidence in the estimated effect and is likely to change the estimate; and (4) very low quality, where any estimated effect is highly uncertain. 

### 2.7. Statistical Analysis

Odds ratio (OR) with 95% confidence interval (CI) was used to calculate the difference for dichotomous outcomes, while the standardized mean difference (SMD) with 95% CI was used for continuous variables. If the studies only reported the median and measure of dispersion, the data were converted to mean and standard deviation assuming a normal distribution, by using two simple formulae. 

We performed a frequentist network meta-analysis of aggregate data to obtain network estimates for the aforementioned outcomes of interest. The model framework used random effects to allow for apparent heterogeneity among studies in treatment comparison effects [25]. We conducted a pairwise meta-analysis to generate direct estimates for outcomes using a random-effects model. 

Transitivity assumption, the distribution of patient, and study characteristics that modify treatment effects (effect modifiers) across treatment comparisons were explored to assess whether these characteristics were sufficiently similar between comparisons. Additionally, we evaluated the incoherence assumption (the statistical disagreement between direct and indirect evidence in a closed loop), locally using a loop-specific approach, and globally using a design by treatment interaction model.

The surface under the cumulative ranking curve (SUCRA) values and rankograms were used to present the hierarchy of interventions for each outcome [26]. SUCRA values show the percentage of effectiveness of each intervention compared to the hypothetically best intervention, which is always the best without uncertainty. The certainty of evidence was assessed using GRADEpro in the Cochrane group. Publication bias was evaluated using a comparison-adjusted funnel plot for network meta-analysis. 

The results were considered statistically significant at a two-sided *p*-value of less than 0.05. All statistical analyses were performed using STATA 14.0 software (StataCorp, College Station, TX, USA).

## 3. Results

### 3.1. Study and Patient Characteristics

We included a total of 15,238 studies according to a prespecified search strategy. Thereafter, 5233 duplicates were removed, and a total of 10,005 studies were left. We excluded 9716 irrelevant studies based on the titles or abstracts, and 289 studies remained. After full-text review, a total of 193 studies were excluded for the following reasons: review articles (*n* = 25), animal studies (*n* = 12), studies with non-relevant interventions (*n* = 24), studies with non-relevant populations (*n* = 62), studies with non-relevant outcomes (*n* = 39), letters (*n* = 4), proceedings (*n* = 4), study protocols (*n* = 11), and non-English (*n* = 4). For the final meta-analysis, a total of 9315 patients were selected from 96 RCTs (Figure 2).

The characteristics of the included studies are shown in Table 1 [8,9,10,11,12,13,14,27,28,29,30,31,32,33,34,35,36,37,38,39,40,41,42,43,44,45,46,47,48,49,50,51,52,53,54,55,56,57,58,59,60,61,62,63,64,65,66,67,68,69,70,71,72,73,74,75,76,77,78,79,80,81,82,83,84,85,86,87,88,89,90,91,92,93,94,95,96,97,98,99,100,101,102,103,104,105,106,107,108,109,110,111,112,113,114,115]. The publication years for the included publications ranged from 2003–2018, and the number of patients ranged from 24 to 600. Seven types of VLs were included in this study, and the number of studies including each VLwere as follows: Airtraq (*n* = 27), Airwayscope (*n* = 16), C-MAC (*n* = 15), CMD (*n* = 8), GlideScope (*n* = 33), King Vision (*n* = 12), McGrath (*n* = 20). These VLs were compared with MCL in 77 studies. Intercomparisons of VLs were conducted in 19 studies. The number of VLS included in each study was as follows: two VLs (*n* = 85), three VLs (*n* = 7), four VLs (*n* = 3), and five VLs (*n* = 1). The top three comparisons for the most frequent VLs were as follows: GlideScope vs. MCL (*n* = 25); Airtraq vs. MCL (*n* = 19); Airwayscope vs. MCL (*n* = 13). The airway circumstances of the study population were classified into the following three categories: all airway (*n* = 23), normal airway (*n* = 33), and difficult airway (*n* = 40). 

### 3.2. Quality Assessment of the Included Studies

The results of the quality assessments of the included studies are presented in Appendix A. All studies showed a high risk of bias in the two domains for blinding of participants and personnel (performance bias) or outcome assessors (detection bias). Most studies showed low or unclear risk of bias in four domains: random sequence generation, allocation concealment, incomplete outcome data, and selective reporting. Only two studies showed a high risk of bias among these four domains. The study by Sarkilar in 2015 was identified as a high risk of bias in random sequence generation due to a time interval of more than one week in the randomization between the MCL and C-MAC groups. Additionally, the study by Cavus in 2011 showed a high risk of bias for incomplete outcome data in the C-MAC group. 

### 3.3. Quantitative Data Synthesis

#### 3.3.1. Intubation Success Rate at First Attempt (Success) 

In the rank analysis using SUCRA, CMD was the overall most successful VL (SUCRA 77.7) and in the context of difficult airway (SUCRA 85.2) status. The pooled success rates of CMD were 90.6% (358/395 patients) in the all airway status category (the range of success rate = 81–100% in eight included studies) [13,29,61,78,89,92,93,103] and 92.7% (178/192 patients) in the difficult airway status category (the range of success rate = 92–95% in three included studies) [29,61,89]. In the normal airway status category, King Vision was the most successful (SUCRA 72.7), and the pooled success rate of King Vision was 92% (169/183 patients; the range of success rate = 68–100% in five included studies) [30,47,48,78,88] (Figure 3, Figure 4 and Figure 5). The success rates in all airway were ranked as follows based on SUCRA values (%); CMD 77.7 (highest), McGrath 76.6, King Vision 67.5, C-MAC 50.9, Airtraq 49.4, GlideScope 48.6, Airwayscope 20.8, MCL 8.6 (lowest). The success rate in normal airway was ranked as follows; King Vision 72.7 (highest), GlideScope 65.9, Airwayscope 60.4, C-MAC 52.0, CMD 48.1, McGrath 42.8, MCL 32.6, Airtraq 25.6 (lowest). The success rates in difficult airway were ranked as follows; CMD 85.2 (highest), McGrath 68.0, C-MAC 64.7, King Vision 58.7, Airtraq 54.9, GlideScope 47.5, Airwayscope 15.9, MCL 4.9 (lowest).

#### 3.3.2. Intubation Time to Confirmation by Capnometry (Speed)

In the rank analysis using SUCRA, C-MAC was the fastest VL in the context of all (SUCRA 84.9) and normal (SUCRA 84.7) airway status. The intubation time of C-MAC ranged from 25 to 32 s (25 ± 7 s vs. 27 ± 7 s vs. 32 ± 6 s; resulted from 144 patients in three included studies) [28,41,116] in all airway status and 27 s (27 ± 7 s; resulting from 39 patients in one included study) [42] in normal airway status category. In the difficult airway condition, the Airwayscope was the fastest (SUCRA 88.7). The intubation time of the Airwayscope was 34 s (34 ± 25 s; resulting from 35 patients in one included study) [67] (Figure 3, Figure 4 and Figure 5). The intubation time in all airway was ranked as follows based on SUCRA values (%); C-MAC 84.9 (highest), Airwayscope 70.4, Airtraq 65.7, King Vision 54.9, CMD 47.4, MCL 31.8, GlideScope 30.9, McGrath 14.1 (lowest). The intubation time in normal airway was ranked as follows; C-MAC 84.7 (highest), GlideScope 59.5, King Vision 54.7, Airtraq 50.6, CMD 49.7, MCL 48.4 (lowest). The intubation time in difficult airway was ranked as follows; Airwayscope 88.7 (highest), Airtraq 79.8, CMD 71.7, King Vision 51.7, C-MAC 41.3, McGrath 40.3, GlideScope 25.8, MCL 0.7 (lowest).

#### 3.3.3. Glottic View (View)

In the rank analysis using SUCRA, Airwayscope showed the best glottic view (SUCRA 84.9) in the normal airway status category (SUCRA 80.2). The pooled rate of good glottic view were 98% (692/699 patients; the range of good glottic view = 86–100% in 13 included studies) [9,35,49,59,64,67,71,72,73,74,99,100,116] in all airway status and 100% (116/116 patients in three included studies) [64,74,116] in the normal airway status category. In the difficult airway status category, McGrath was best in glottic view (SUCRA 84.9), and the pooled rate of good glottic view was 98% (148/151 patients resulted from four included studies) [50,101,114,115] (Figure 3, Figure 4 and Figure 5). When sorting by rank, glottic view ranking in all airway based on SUCRA values, from best to worst, were as follows; Airwayscope 84.9, McGrath 78.3, King Vision 61.6, Airtraq 59.8, CMD 46.6, GlideScope 42.6, C-MAC 24.8, MCL 1.5. Glottic view ranking in normal airway based on SUCRA values, from best to worst, were as follows; Airwayscope 80.2, CMD 68.2, McGrath 56.2, GlideScope 47.9 = King Vision 47.9, Airtraq 46.5, C-MAC 40.2, MCL 12.9. Glottic view ranking in difficult airway based on SUCRA values, from best to worst, were as follows; McGrath 80.1, King Vision 75.3, Airtraq 65.7, Airwayscope 53.4, GlideScope 47.3, C-MAC 27.7, MCL 0.4.

#### 3.3.4. Sore Throat within 24 h after Extubation (Safety) 

In the rank analysis using SUCRA, CMD showed the lowest incidence of sore throat within 24 h after extubation in all (SUCRA 99.9) and normal (SUCRA 99.3) airway status categories. The pooled incidence of sore throat for CMD was 0% (0/65 patients resulted from one included study) in both airway status categories [103]. In the difficult airway status category, Airtraq also showed a low incidence of sore throat (SUCRA 83.5). The pooled incidence rate of sore throat in Airtraq was 12.6% (26/206 patients; the range of incidence = 0–17.5% in four included studies) [39,60,76,102] (Figure 3, Figure 4 and Figure 5). When sorting by rank, safety ranking in all airway based on SUCRA values were as follows; CMD 99.9, Airtraq 66.1, King Vision 58.0, Airwayscope 52.8, McGrath 46.1, MCL 32.3, GlideScope 24.1, C-MAC 20.8. Safety ranking in normal airway were as follows; CMD 99.3, Airwayscope 71.9, McGrath 60.6, Airtraq 55.9, King Vision 47.6, C-MAC 30.2, GlideScope 19.9, MCL 14.7. Safety ranking in difficult airway were as follows; Airtraq 83.5, King Vision 58.2, GlideScope 53.1, MCL 48.8, C-MAC 38.2, McGrath 38.1, Airwayscope 30.1.

### 3.4. Quality Evidence in GRADE Assessment

The evidence level of each comparison between VL and MCL or intercomparison of the VLs is fully described in Appendix A. We extracted and summarized the comparisons of all moderate certainty of evidence assessed by the GRADE tool in Table 2. 

For intubation success, five VLs (Airtraq, CMD, GlideScope, King Vision, and McGrath) were more successful than MCL. Moderate evidence did not exist in the intercomparison of VLs. For intubation time, four VLs (Airwayscope, C-MAC, GlideScope, and McGrath) were faster than MCL. In the intercomparison of VLs, the Airtraq was faster than other VLs (CMD, GlideScope, King Vision in difficult airway, and McGrath). The Airwayscope was also faster than GlideScope. However, GlideScope was faster than McGrath and King Vision in the normal airway category, whereas it was slower than McGrath in the difficult airway category. For the glottic view, Airtraq and Airwayscope showed better glottic views than MCL. No moderate evidence existed in the intercomparison of VLs for the glottic view. For safety after extubation, three VLs (Airtraq, C-MAC, and CMD) showed a lower incidence of sore throat than MCL. Moreover, two VLs (Airtraq and McGrath) showed a lower incidence of sore throat than GlideScope in the intercomparisons of VLs.

### 3.5. Publication Bias

In the comparison-adjusted funnel plot, most funnel plots showed symmetry for the success, speed, view, and safety in three airway status categories (all vs. normal vs. difficult). Asymmetry was only observed in sore throat (safety) in all, normal, and difficult airway categories, which suggested the presence of small-study effects (Appendix A).

## 4. Discussion

This systematic review and network meta-analysis demonstrated that the CMD was relatively successful compared with other VLs and MCL for the tracheal intubation undergoing general anesthesia in all airway circumstances. Additionally, while the KV was more successful in normal airway circumstances, the CMD was more successful in difficult airway circumstances compared with other VLs and MCL. Other VLs such as MG, GVL, AWS, and CM were top three ranked VLs for intubation success in all, normal and difficult airway circumstances. The comparisons of intubation success between VLs and MCL showed moderate certainty of evidence level, whereas the intercomparisons of VLs showed low certainty evidence.

Previous meta-analyses have provided limited evidence for the usefulness of VLs compared with MCL. De jong et al. only identified the usefulness of VLs compared with DL in critical care settings through a 2014 meta-analysis [117]. In both emergency and critical care settings, the meta-analysis for seven RCTs reported that the first-pass intubation success was not significantly improved by VLs compared with DL [118]. In the 2016 Cochrane systematic review by Lewis, the 64 included studies were composed of 61 elective surgery patients and three patients in emergency settings [18]. However, this study found that VLs mostly showed better performance than DL. We hypothesized that more widespread usage and the availability of VLs in training programs will lead to improved VL performance in clinical settings. Although this study demonstrated that VLs might reduce failed intubation in difficult airways, no evidence indicated that the use of VLs affected the time required for intubation. No evidence was provided for the outcomes for intubation in the intercomparison of VLs in these meta-analyses.

To obtain consistent results and minimize the heterogeneity in the comparison of laryngoscopes, we categorized airway circumstances (normal vs. difficult) and only included studies for intubation by experienced anesthetists in elective surgery. In emergent or critical care settings, some concerns might be raised for inaccurate or rough evaluation of airway status. Furthermore, the urgency of the situation might significantly affect the intubation time or success, especially in cases of difficult airway.

The highest-ranked VLs for outcomes were CMD (success, safety), King Vision (success), Airwayscope (speed and view), C-MAC (speed), McGrath (view), and Airtraq (safety). In this study, CMD, C-MAC, and McGrath were categorized as non-channeled VLs, whereas King Vision, Airwayscope, and Airtraq were categorized as channeled VLs. Through these results, we realized that no absolute superiority existed between non-channeled and channeled VLs. These results also inspired that the most appropriate VL should be clinically decided by considering airway circumstances and the characteristics of VLs. 

The included VLs have slightly different characteristics in the blade or video screen, to aid intubation. Most channeled VLs such as Airtraq, Airwayscope, and King Vision have angulated disposable blades and direct screens combined with handles [30,68,73]. Because Airtraq has an exaggerated blade curvature (90°), a view of the glottis can be provided with minimal need for airway optimisation manoeuvres such as hyperextension [68]. Compared with Airtraq, King Vision has a wider field of view (160° vs. 80°, respectively) with a similar blade curvature [30]. Non-channeled VLs such as GlideScope, C-MAC, and CMD require the use of a stylet and, with the exception of McGrath, have indirect screens [34,43,119]. GlideScope and C-MAC have a lesser angulated blade (60° vs. 80°, respectively) than Airtraq and King Vision [28,34]. In particular, CMD has an exaggerated curvature of the distal end of the blade, which faces markedly upward [119]. As a result of the curvature of the blade components, anesthetists require less cervical spine movement. 

In the analysis of intubation success in a difficult airway, we additionally found that the different experiences for specific VLs might act as a confounding factor even though experienced anesthetists were enrolled. Although the pooling success rate of intubation using King Vision was 92% in the normal airway category (169/183; 68–100% in five included studies) [30,47,48,78,88], the study by Abdulmohsen (2016) reported a very low success rate for King Vision (68%) as well as a significantly lesser performance for King Vision and Airtraq (intubation >15 times: King Vision 3/25 vs. Airtraq 0/25 vs. GlideScope 20/25 vs. MCL 25/25) [30]. Among the other four included studies, both Mendonca 2018 and Reena 2019 showed sufficient experience for intubation (King Vision intubation > 50 times) [78,88]. However, two other studies by El-Tahan reported unclear or insufficient experience for intubation (median King Vision intubation <12 times), even if the reported success rate was 100% [47,48]. Thus, the experience of intubation for King Vision might play a role as a confounder for the success rate. However, in the analysis for intubation experience in Appendix A, the included studies did not report outcomes by consistent manner enough to perform meta-regression. Therefore, we only summarized the information of intubation experiences and intubators. This information suggested that most investigators have enough experiences for MCL, however, the experience for VLs was relatively insufficient compared with MCL. The difference of intubation experience between MCL and VLs should be considered in the evaluation of intubation performances of VLs. 

This study has some limitations. First, some important confounders might affect the outcomes of this study. Following factors might affect the intubation performance: the different intubation experience (resident vs. attending level; working history; total intubation attempts of each device), the systemic diseases of patients such as American Society of Anesthesiologists score, the hemodynamic characteristics of patients, the use of sedatives or muscle relaxants. The size and type of ETT (double lumen vs. single lumen) might also be a contributing factor for sore throat. Non-channeled video laryngoscopes (VLs) performed better in intubation than channeled VLs, videostylets, and direct laryngoscopes, according to a prior network meta-analysis by Kim et al. 2020 [120]. We concentrated on obtaining more general findings, regardless of the lumen type or ETT size. ETT’s increased size could be linked to an increase in sore throat. However, the size of ETT is an ordinary variable with a narrow range, making it an insufficient variable for meta-regression. As good eye–hand coordination is necessary for intubation using non-channeled VL compared to channeled VL, the use of a stylet in non-channeled VL may be linked to an increased risk of sore throat [121]. By causing distinct hemodynamic changes during intubation, sufficient muscular relaxation has a major impact on the ease of laryngoscopy and intubation success. Depolarizing or non-depolarizing muscle relaxants may impact the results. We summarized this information in Appendix A. Further sensitivity or subgroup analysis for these possible confounders was difficult to perform in the network meta-analysis because this information was not reported by consistent manner to perform further analyses. Second, small-study effects for sore throat safety were suggested in the comparison-adjusted funnel plot. This did not indicate the presence of publication bias. It was needed to interpret carefully not to overestimate the effect of the intervention. Third, certain critical criteria for defining a difficult airway were missed. Patients’ histories, complaints, clinical examinations, and lastly investigative findings all fall under this category. As a result, results from meta-analyses that have been pooled for restricted information cannot adequately reflect the clinical setting. Fourth, there is no absolute distinction between channeled (without stylet) and non-channeled (with stylet) VLs. Intubation with the ATQ, for example, may require the use of a gum elastic bougie, but intubation with the GVL, CM, or MG does not. Fifth, some risk factors that were left out of the study could be linked to the occurrence or duration of sore throat. In the literature search and full text review, the majority of the studies just stated “elective surgery” rather than addressing specific surgeries. As a result, we were unable to determine whether upper airway surgery was linked to sore throat. Other details, such as a history of previous difficult intubation or a non-infectious or infective sore throat, were not clearly indicated. Sixth, the number of attempts at intubation and cuff pressure were significant factors in the rise in intubation-related complications. We could not find any data on cuff pressure in any of the research we looked at. According to the complete text review, the number of intubation attempts ranged from one to five times. For meta-analysis, however, it was insufficient and heterogeneous to synthesize. As a result, we only used the first attempt at intubation as a primary outcome because the data was sufficient and homogeneous. Seventh, several intubation-related complications, as well as sore throat, were thoroughly investigated. Other mechanical complications have already been collected, such as mucosal bleeding and dental injury. As the data were insufficient and heterogeneous, we were unable to synthesize it in the network meta-analysis.

In conclusion, CMD and King Vision could be relatively successful VLs for tracheal intubation in the comparisons of MCL and other VLs under general anesthesia. The comparisons of intubation success between VLs and MCL showed moderate certainty of evidence level, whereas the intercomparisons of VLs showed low certainty evidence. 

## Figures and Tables

**Figure 1 jpm-12-00363-f001:**
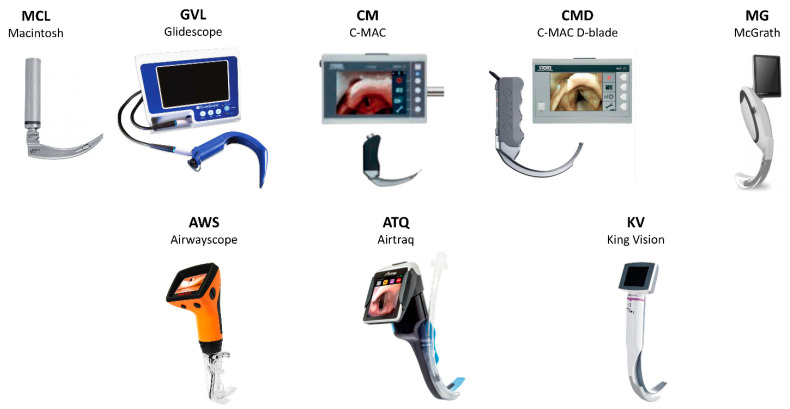
Type of videolaryngoscope. MCL, Macintosh direct laryngoscope; ATQ, Airtraq video laryngoscope; AWS, Pentax Airwayscope; CM, C-MAC video laryngoscope; CMD, C-MAC D-Blade video laryngoscope; GVL, GlideScope; KV, King Vision video laryngoscope; MG, McGrath video laryngoscope.

**Figure 2 jpm-12-00363-f002:**
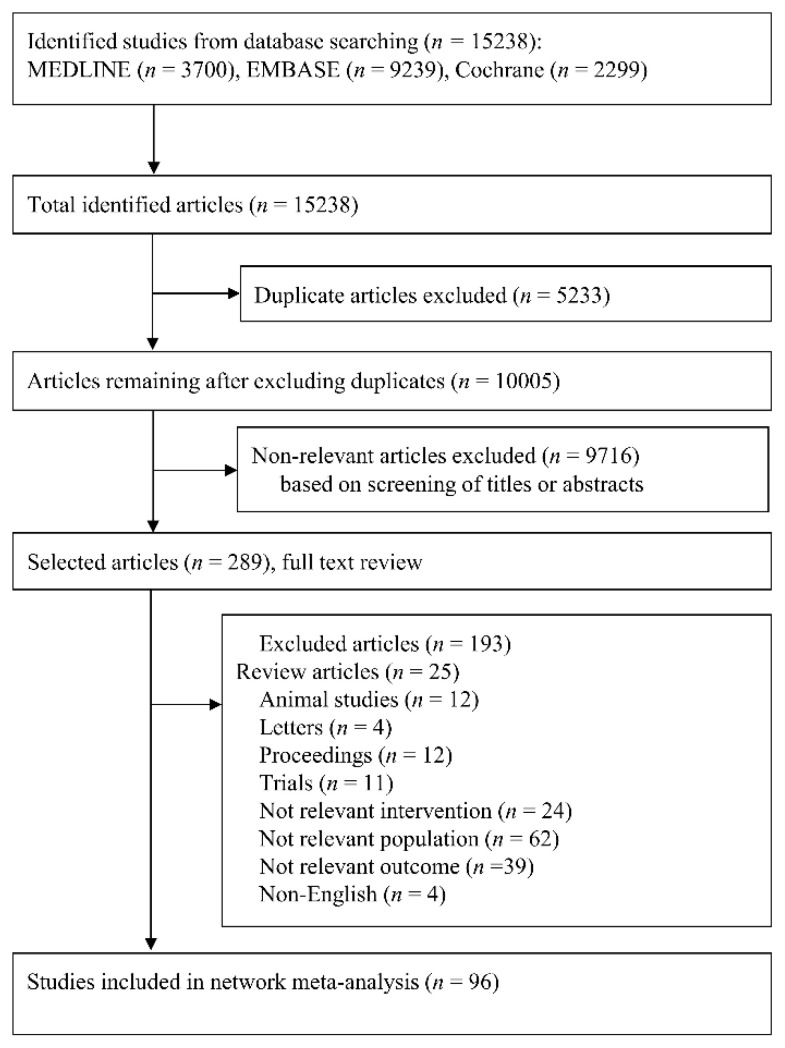
Preferred Reporting Items for Systematic Reviews and Meta-Analyses (PRISMA) flowchart of search strategy and study selection.

**Figure 3 jpm-12-00363-f003:**
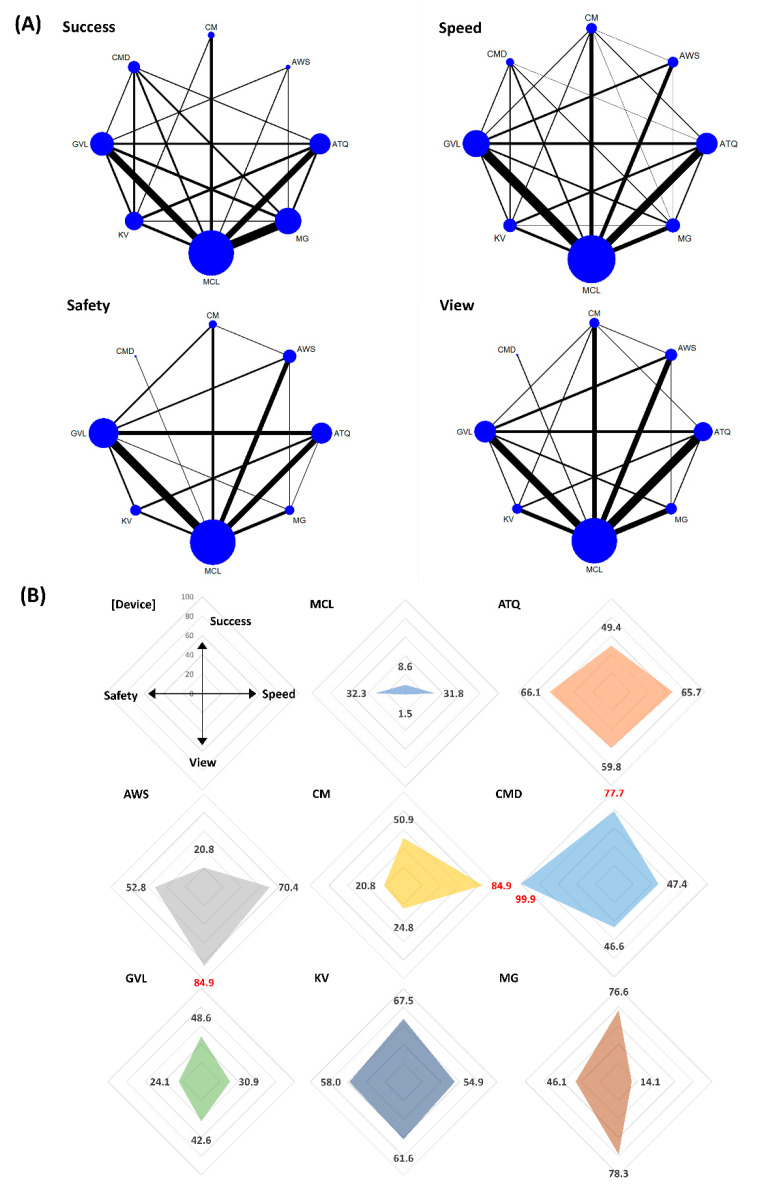
Comparison of efficacy and safety for intubation using video laryngoscopes in all circumstances of airway. (**A**) Network maps for the successful, speedy and good glottic view at first intubation attempt (efficacy), and sore throat within 24 h after extubation (safety). (**B**) Radar charts using the surface under the cumulative ranking (SUCRA) for efficacy and safety in each video laryngoscope. The highest value of SUCRA for each outcome was marked in red. MCL, Macintosh direct laryngoscope; ATQ, Airtraq video laryngoscope; AWS, Pentax Airwayscope; CM, C-MAC video laryngoscope; CMD, C-MAC D-Blade video laryngoscope; GVL, GlideScope; KV, King Vision video laryngoscope; MG, McGrath video laryngoscope.

**Figure 4 jpm-12-00363-f004:**
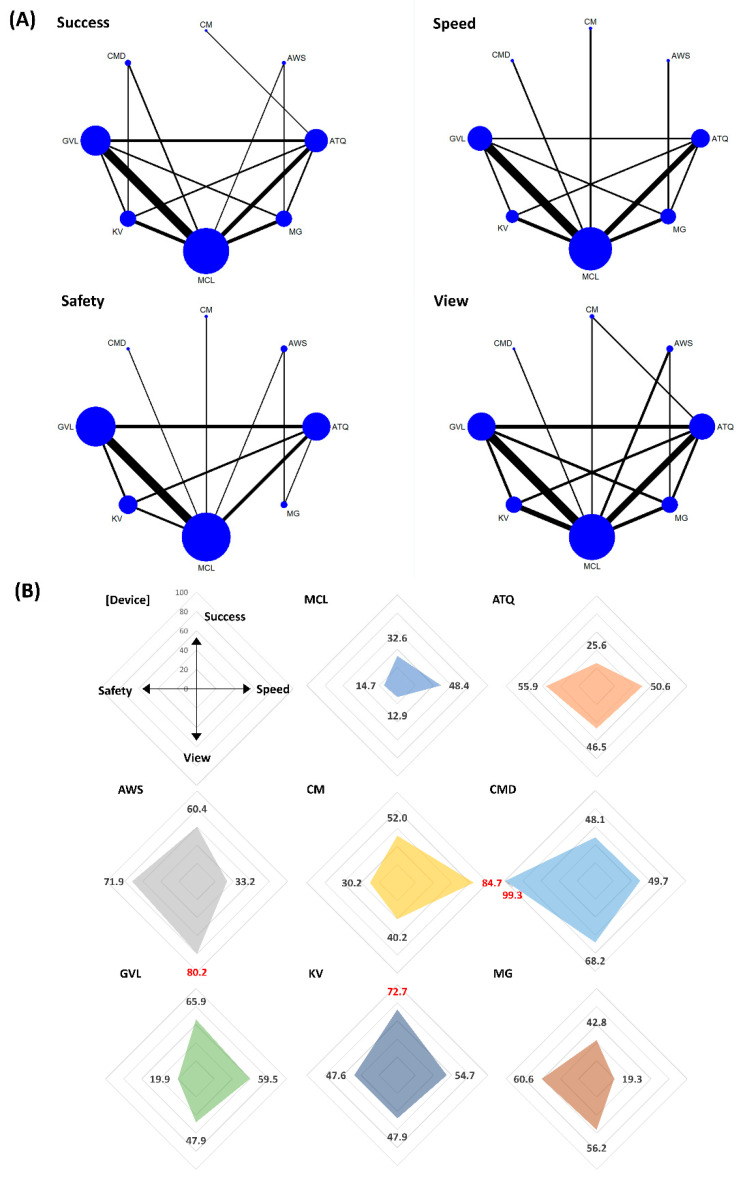
Comparison of efficacy and safety for intubation using video laryngoscopes in normal airway. (**A**) Network maps for the successful, speedy and good glottic view at first intubation attempt (efficacy), and sore throat within 24 h after extubation (safety). (**B**) Radar charts using the surface under the cumulative ranking (SUCRA) for efficacy and safety in each video laryngoscope. The highest value of SUCRA for each outcome was marked in red. MCL, Macintosh direct laryngoscope; ATQ, Airtraq video laryngoscope; AWS, Pentax Airwayscope; CM, C-MAC video laryngoscope; CMD, C-MAC D-Blade video laryngoscope; GVL, GlideScope; KV, King Vision video laryngoscope; MG, McGrath video laryngoscope.

**Figure 5 jpm-12-00363-f005:**
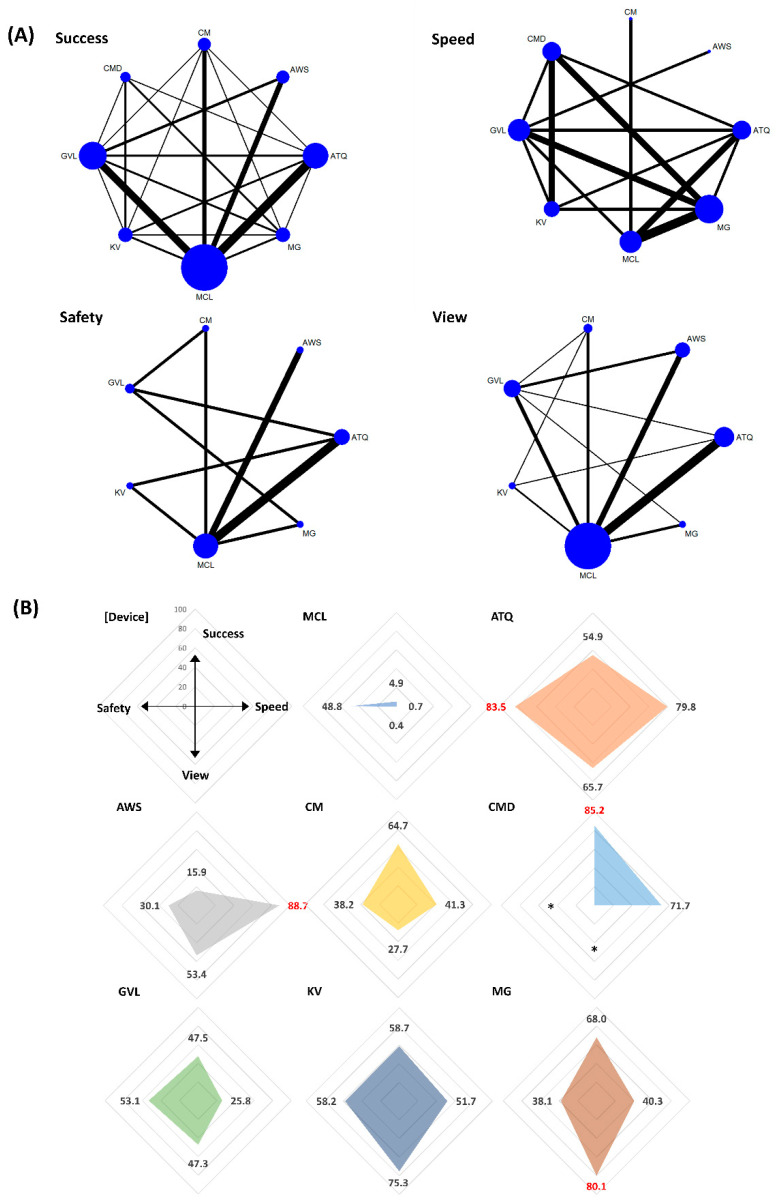
Comparison of efficacy and safety for intubation using video laryngoscopes in difficult airway. (**A**) Network maps for the successful, speedy and good glottic view at first intubation attempt (efficacy), and sore throat within 24 h after extubation (safety). (**B**) Radar charts using the surface under the cumulative ranking (SUCRA) for efficacy and safety in each video laryngoscope. The highest value of SUCRA for each outcome was marked in red. MCL, Macintosh direct laryngoscope; ATQ, Airtraq video laryngoscope; AWS, Pentax Airwayscope; CM, C-MAC video laryngoscope; CMD, C-MAC D-Blade video laryngoscope; GVL, GlideScope; KV, King Vision video laryngoscope; MG, McGrath video laryngoscope. * lack of study.

**Table 1 jpm-12-00363-t001:** Characteristics of the included randomized controlled study.

Study, Year	Country	Patients Number	Airway Status	Devices	Tracheal Tube (Size; mm or Fr)	Success Rate (*n*)	Intubation Time (s)	Good Glottic View (*n*)	Sore Throat (*n*)
Abdallah, 2011	USA	99	Difficult	MCL AWS	ETT (7/7.5 mm)	45/49 43/50	N/A	38/49 43/50	16/50 16/49
Abdulmohsen, 2016	Saudi Arabia	86	Normal	MCL ATQ GVL KV	ETT (7/8 mm)	16/22 9/21 15/21 15/22	35.1 ± 8.61 56.4 ± 6.02 41 ± 6.41 47.5 ± 8.94	19/22 19/21/ 19/21 22/22	19/22 6/21 16/21 11/22
Ahmed, 2017	India	60	Normal	ATQ CM	ETT	27/30 28/30	N/A	28/30 28/30	N/A
Akbar, 2015	Malaysia	90	Difficult	MCL CM	ETT (7/7.5/8 mm)	39/45 44/45	38.8 ± 8.9 32.7 ± 6.8	42/45 43/45	N/A
Akbas, 2019	Turkey	80	Difficult	CMD MG	ETT (7/7.5 mm)	37/40 37/40	38.65 ± 17.57 55.2 ± 6.32	N/A	N/A
Ali, 2012	India	50	All	MCL ATQ	ETT	16/25 22/25	36 ± 16 48 ± 18	N/A	N/A
Ali, 2015	India	50	All	ATQ KV	ETT	21/25 24/25	38 ± 18 26 ± 11	N/A	N/A
Ali, 2017	India	60	Difficult	MCL KV	ETT (7/8 mm)	27/30 29/30	N/A	24/30 28/30	N/A
Amini, 2015	Iran	70	Normal	MCL GVL	ETT	N/A	9.3 ± 1.4 10.6 ± 1.7	N/A	15/35 13/35
Ander, 2017	Sweden	78	Difficult	MCL CM	ETT (6/7/8 mm)	34/39 39/39	N/A	N/A	4/36 8/39
Aoi, 2010	Japan	36	Difficult	MCL AWS	ETT	14/18 14/18	N/A	11/18 18/18	4/18 8/18
Aqil, 2016	Saudi Arabia	80	Normal	MCL GVL	ETT (7/8 mm)	33/40 34/40	41.3 ± 15.2 32.9 ± 8.6	36/40 39/40	N/A
Aqil, 2017	Saudi Arabia	140	Normal	MCL GVL	ETT (7/8 mm)	55/70 64/70	N/A	56/70 67/70	15/70 7/70
Arici, 2014	Turkey	80	All	MCL MG	ETT	40/40 40/40	32.2 ± 6.58 47.25 ± 14.92	40/40 38/40	N/A
Bakshi, 2019	India	74	Normal	MCL MG	ETT* (35/37/39 Fr)	35/3736/37	56.6 ± 14 64.4 ± 24	36/37 37/37	N/A
Belze, 2017	France	72	Difficult	ATQ GVL	ETT* (35/37/39/41 Fr)	28/36 29/36	N/A	36/36 33/36	34/36 33/36
Bhandari, 2013	India	80	Normal	MCL ATQ	ETT	38/40 40/40	29 ± 5.4 18 ± 2.6	34/36 36/36	36/36 34/36
Bilehjani, 2009	Iran	80	Normal	MCL GVL	ETT	29/40 35/40	N/A	N/A	N/A
Blajic, 2019	Slovenia	178	All	MCL CM KV	ETT (6.5 mm)	56/59 59/60 56/59	29 ± 14 25 ± 7 29 ± 9	52/59 55/60 57/59	N/A
Bruck, 2015	Germany	56	Difficult	CM GVL	ETT (7.5/8 mm)	15/26 28/30	N/A	26/26 30/30	9/26 12/30
Caparlar, 2019	Turkey	78	Normal	MCL CM	ETT	N/A	41.49 ± 10.3 27.74 ± 7.2	39/39 39/39	0/39 0/39
Cavus, 2011	Germany	87	All	MCL CM	ETT	48/50 27/37	N/A	N/A	N/A
Chalkeidis, 2010	Greece	63	All	MCL ATQ	ETT	27/28 31/35	N/A	N/A	N/A
Chandrashek-araiah, 2017	Bahrain	60	Difficult	MCL CM	ETT	N/A	N/A	26/30 28/30	N/A
Dhonneur, 2009	France	212	Difficult	MCL ATQ	ETT	N/A	N/A	90/106 106/106	N/A
ElTahan, 2017	Saudi Arabia	29	Normal	MCL KV	ETT	29/29 29/29	N/A	N/A	N/A
ElTahan, 2018	Saudi Arabia	133	Normal	MCL ATQ GVL KV	ETT* (35/37/39 Fr)	32/32 33/35 34/34 32/32	N/A	28/32 30/35 29/34 25/32	2/32 1/35 29/34 2/32
Enomoto, 2008	Japan	203	Difficult	MCL AWS	ETT (7/8 mm)	N/A	N/A	181/203 203/203	N/A
Foulds, 2016	England	49	Difficult	MCL MG	ETT	18/25 24/24	95.3 ± 55.2 55 ± 18.5	28/49 0/49	N/A
Gupta, 2013	India	60	Difficult	MCL CM	ETT (7.5/8.5 mm)	28/30 30/30	N/A	12/30 26/30	N/A
Hosalli, 2017	India	60	Difficult	MCL ATQ	ETT	23/30 27/30	N/A	27/30 30/30	N/A
Hsu, 2012	Taiwan	60	Normal	MCL GVL	ETT* (35/37 Fr)	26/30 30/30	62.5 ± 29.7 45.6 ± 10.7	N/A	12/30 6/30
Hu, 2017	China	196	Difficult	MCL GVL	ETT (6/6.5/7 mm)	95/96 100/100	N/A	46/96 88/100	N/A
Ilyas, 2014	Australia	128	Difficult	MCL MG	ETT (7/7.5/8/8.5 mm)	N/A	N/A	N/A	39/64 40/64
Jafra, 2018	India	200	Normal	MCL GVL	ETT	100/100 100/100	N/A	86/100 77/100	0/100 1/100
Jeon, 2011	South Korea	56	Normal	GVL MG	ETT (7/7.5 mm)	25/28 27/28	41.6 ± 10.7 56.5 ± 23.2	28/28 28/28	N/A
Kido, 2015	Japan	50	All	MCL MG	ETT* (32/35/37 Fr)	16/25 24/25	20.8 ± 5.9 17.1 ± 4.6	20/25 25/25	14/25 7/25
Kim, 2013	South Korea	45	All	MCL AWS	ETT (7/7.5/8 mm)	19/23 22/22	24.3 ± 16.6 12.9 ± 6	8/23 22/22	N/A
Kleine-Brueg-geney, 2016	Switzerland	600	Difficult	ATQ CMD GVL KV MG	ETT (6.5/7.5 mm)	102/120 114/120 102/120 104/120 117/120	47.7 ± 18 62 ± 30 68.7 ± 37.5 61 ± 24 57.3 ± 26.2	N/A	N/A
Kleine-Brueg-geney, 2017	Switzerland	360	Difficult	MCL ATQ KV	ETT (6.5/7.5 mm)	53/120 98/120 108/120	N/A	7/120 107/120 116/120	28/120 21/120 26/120
Lange, 2009	Germany	60	All	ATQ GVL	ETT	28/30 29/30	N/A	28/30 30/30	15/30 12/30
Lee, 2012	Netherlands	75	Normal	MCL GVL MG	ETT	21/25 25/25 25/25	N/A	21/25 25/25 25/25	N/A
Lee, 2013	South Korea	40	Normal	MCL AWS	ETT (6.5/7.5 mm)	N/A	N/A	N/A	0/20 0/20
Lee, 2017	South Korea	140	Normal	AWS MG	ETT (7 mm)	70/70 70/70	30.3 ± 5.3 31.3 ± 6.1	70/70 67/70	10/70 17/70
Lim, 2005	Singapore	60	Difficult	MCL GVL	ETT (7 mm)	26/30 28/30	56.2 ± 27 41.8 ± 20	12/30 28/30	N/A
Liu 2009	Japan	70	Difficult	AWS GVL	ETT	34/35 29/35	34.2 ± 25.1 71.9 ± 47.9	35/35 35/35	N/A
Maharaj, 2006	Ireland	60	Normal	MCL ATQ	ETT	29/30 30/30	N/A	29/30 30/30	N/A
Maharaj, 2007	Ireland	40	Difficult	MCL ATQ	ETT	19/20 20/20	N/A	13/20 20/20	N/A
Maharaj, 2008	Ireland	40	Difficult	MCL ATQ	ETT	13/20 19/20	N/A	3/20 20/20	N/A
Malik, 2008	Ireland	90	Difficult	MCL AWS GVL	ETT (7.5/8.5 mm)	26/30 27/30 28/30	N/A	25/30 30/30 30/30	N/A
Malik1, 2009	Ireland	75	Difficult	MCL AWS GVL	ETT (7.5/8.5 mm)	17/25 18/25 22/25	N/A	17/25 25/25 25/25	N/A
Malik2, 2009	Ireland	60	Difficult	MCL AWS	ETT (7.5/8 mm)	29/30 28/30	N/A	25/30 30/30	N/A
Maruyama, 2008	Japan	24	Normal	MCL AWS	ETT	N/A	N/A	12/12 12/12	N/A
Maruyama, 2011	Japan	68	Normal	MCL AWS	ETT (7 mm)	32/34 33/34	N/A	31/34 34/34	N/A
Mathew, 2018	India	60	Difficult	MCL ATQ	ETT (7/7.5/8/8.5 mm)	29/30 27/30	N/A	25/30 21/30	26/30 27/30
McElwain, 2011	Ireland	90	Difficult	MCL ATQ CM	ETT	25/31 28/29 26/30	N/A	N/A	N/A
Mendonca, 2018	England	100	Normal	KV CMD	ETT	47/50 48/50	N/A	N/A	N/A
Najafi, 2014	Iran	300	All	MCL GVL	ETT (7.5/8 mm)	N/A	N/A	N/A	81/150 34/150
Nandakumar, 2018	India	30	Difficult	MCL GVL	ETT (7/7.5/8/8.5 mm)	13/15 11/15	N/A	N/A	N/A
Ndoko, 2008	France	106	Difficult	MCL ATQ	ETT	49/53 53/53	N/A	42/53 53/53	N/A
Ng, 2012	Australia	130	Difficult	MG CM	ETT (7/7.5/8/8.5 mm)	45/65 58/65	N/A	N/A	N/A
Ninan, 2016	India	60	All	MCL CM	ETT	30/30 30/30	N/A	20/30 21/30	N/A
Nishikawa, 2009	Japan	40	All	MCL AWS	ETT (7/8 mm)	N/A	N/A	N/A	6/20 2/20
Parasa, 2016	India	60	All	MCL GVL	ETT (7/8 mm)	30/30 24/30	27.77 ± 5.12 45.70 ± 11.65	26/30 29/30	8/30 10/30
Pazur, 2016	Croatia	52	Normal	MCL CMD	ETT (7.5/8.5 mm)	26/26 26/26	34.3 ± 15.1 33.6 ± 16.7	24/26 26/26	N/A
Ranieri, 2012	Brazil	132	Difficult	MCL ATQ	ETT (7.5/8.5 mm)	54/64 68/68	37 ± 23 14 ± 3	57/64 68/68	N/A
Raza, 2017	India	60	Normal	MG ATQ	ETT	25/30 27/30	N/A	30/30 30/30	N/A
Reena, 2019	India	100	Normal	MCL KV	ETT (7/8 mm)	37/50 46/50	40.3 ± 14.4 28.7 ± 10.6	41/50 48/50	N/A
Russell, 2013	Canada	70	Normal	MCL GVL	ETT*	32/35 29/35	N/A	N/A	2/35 5/35
Sahajanandan, 2019	India	63	Difficult	KV CMD	ETT	23/31 27/32	50.04 ± 24.17 46.93 ± 26.54	N/A	N/A
Sargin, 2016	Turkey	100	Normal	MCL MG	ETT (7/7.5 mm)	50/50 50/50	N/A	N/A	N/A
Sarkilar, 2015	Turkey	110	All	MCL CM	ETT (8/9 mm)	55/55 55/55	N/A	49/55 53/55	N/A
Serocki, 2013	Germany	96	All	MCL CMD GVL	ETT (7/8 mm)	27/32 27/32 29/32	N/A	N/A	N/A
Shah, 2016	India	60	All	MCL CMD	ETT* (35/37/39 Fr)	16/30 26/30	81.41 ± 19.7 88.75 ± 14.33	19/30 26/30	N/A
Shravanalakshmi, 2017	India	90	Difficult	CM KV	ETT (7/8 mm)	45/45 42/45	N/A	45/45 45/45	N/A
Siddiqui, 2009	Canada	40	Normal	MCL GVL	ETT (7/8 mm)	N/A	N/A	N/A	2/20 4/20
Sun, 2005	Canada	200	All	MCL GVL	ETT	97/100 94/100	N/A	82/100 85/100	N/A
Taylor, 2013	Canada	88	All	MCL MG	ETT (7/7.5 mm)	26/44 44/44	21.7 ± 9.4 35.8 ± 20.4	1/44 44/44	8/44 5/44
Tempe, 2016	India	39	Normal	MCL MG	ETT	17/19 14/20	24.53 ± 10.9 52.3 ± 28.1	18/19 20/20	N/A
Teoh, 2009	Singapore	140	All	AWS GVL	ETT (7 mm)	61/70 64/70	N/A	70/70 70/70	0/70 13/70
Teoh, 2010	Singapore	400	All	MCL AWS CM GVL	ETT (7 mm)	98/100 95/100 93/100 91/100	N/A	95/100 100/100 98/100 99/100	3/100 1/100 8/100 15/100
Toker, 2019	Turkey	100	Difficult	MCL MG	ETT	N/A	40.1 ± 5.4 34.7 ± 5.2	37/50 48/50	N/A
Tolon, 2012	Egypt	40	Difficult	MCL ATQ	ETT	20/20 20/20	48.75 ± 21.57 34.3 ± 12.27	17/20 20/20	1/20 0/20
Tosh, 2018	India	130	Normal	MCL CMD	ETT (7/8 mm)	54/65 53/65	N/A	N/A	44/65 0/65
Turkstra, 2016	Canada	160	Normal	MCL GVL	ETT (7/7.5/8/8.5 mm)	76/80 74/80	48.2 ± 17.1 51.5 ± 21.8	79/80 78/80	N/A
Varsha, 2019	India	70	Normal	MCL ATQ	ETT (7/7.5/8/8.5 mm)	33/35 35/35	31 ± 37.1 27 ± 29.37	24/35 35/35	N/A
Vijayakumar, 2016	India	90	Difficult	MCL ATQ	ETT (7/8 mm)	N/A	N/A	33/45 45/45	N/A
Wan, 2016	China	87	Normal	ATQ MG	ETT* (35/37/39 Fr)	40/43 42/44	28.6 ± 13.6 39.9 ± 9.1	43/43 44/44	8/43 5/44
Wasem, 2013	Germany	60	All	MCL ATQ	ETT* (35/37/39/41 Fr)	26/30 28/30	N/A	30/30 30/30	6/30 7/30
Wasinwong, 2017	Thailand	46	Difficult	MCL GVL	ETT (7.5/8 mm)	21/23 23/23	N/A	N/A	N/A
Woo, 2012	South Korea	159	All	MCL AWS	ETT (7/8 mm)	50/109 50/50	N/A	N/A	26/109 29/50
Xue, 2007	China	57	Difficult	MCL GVL	ETT (7/7.5 mm)	27/27 28/30	N/A	N/A	N/A
Yao, 2015	China	96	All	MCL MG	ETT* (35/37/39 Fr)	48/48 48/48	24.3 ± 7.1 29.7 ± 10.5	48/48 48/48	6/48 8/48
Yi, 2015	China	70	Normal	ATQ GVL	ETT* (35/37/39 Fr)	33/35 34/35	N/A	28/35 35/35	6/35 8/35
Yoo, 2018	Korea	44	Difficult	MCL MG	ETT* (35/37 Fr)	17/22 21/22	52.7 ± 11.1 45 ± 11.1	12/22 21/22	N/A
Yumul, 2016	USA	60	Difficult	GVL MG	ETT	28/30 21/30	69 ± 34 62 ± 31	28/30 30/30	7/30 11/30

ETT, Endotracheal tube; ETT*, the use of double-lumen endotracheal tube in open thoracic surgery; Fr, French(scale); N/A, Not applicable; MCL, Macintosh laryngoscope; ATQ, Airtraq; AWS, Airwayscope; CM, C-MAC; CMD, C-MAC d-blade; GVL, GlideScope; KV, King Vision; MG, McGrath.

**Table 2 jpm-12-00363-t002:** Moderate certainty of evidence of comparison between VL and MCL or intercomparison of VLs.

	Participants (Studies)	Intervention	Comparison	Airway Status	Study Event Rates (%)		Relative Effect (95% CI)	Anticipated Absolute Effects	
					Comparison	Intervention		Risk with Comparison	Risk Difference with Intervention
Success	778 (9 RCTs)	ATQ	MCL	Difficult	285/388 (73.5%)	360/390 (92.3%)	OR 3.06 (1.39 to 6.72)	735 per 1000	160 more per 1000 (from 59 more to 214 more)
	684 (7 RCTs)	KV	MCL	All	250/342 (73.1%)	315/342 (92.1%)	OR 2.01 (1.01 to 4.01)	731 per 1000	114 more per 1000 (from 2 more to 185 more)
	300 (2 RCTs)	KV	MCL	Difficult	80/150 (53.3%)	137/150 (91.3%)	OR 3.31 (1.14 to 9.56)	533 per 1000	258 more per 1000 (from 32 more to 383 more)
	306 (4 RCTs)	CMD	MCL	All	123/153 (80.4%)	132/153 (86.3%)	OR 2.36 (1.06 to 5.26)	804 per 1000	102 more per 1000 (from 9 more to 152 more)
	93 (2 RCTs)	MCL	MG	Difficult	45/46 (97.8%)	35/47 (74.5%)	OR 0.25 (0.08 to 0.76)	978 per 1000	60 fewer per 1000 (from 196 fewer to 7 fewer)
	1972 (21 RCTs)	GVL	MCL	All	918/990 (92.7%)	897/982 (91.3%)	OR 1.59 (1.00 to 2.54)	927 per 1000	26 more per 1000 (from 0 fewer to 43 more)
	504 (7 RCTs)	GVL	MCL	Difficult	237/251 (94.4%)	240/253 (94.9%)	OR 2.68 (1.23 to 5.81)	944 per 1000	34 more per 1000 (from 10 more to 46 more)
Intubation time	240 (1 RCT)	ATQ	CMD	All	120	120	-	-	SMD 0.58 SD lower (0.83 lower to 0.32 lower)
	240 (1 RCT)	ATQ	CMD	Difficult	120	120	-	-	SMD 0.58 SD lower (0.83 lower to 0.32 lower)
	43 (1 RCT)	ATQ	KV	Normal	22	21	-	-	SMD 1.14 SD higher (0.49 higher to 1.79 higher)
	240 (1 RCT)	ATQ	KV	Difficult	120	120	-	-	SMD 0.62 SD lower (0.88 lower to 0.37 lower)
	42 (1 RCT)	ATQ	GVL	Normal	21	21	-	-	SMD 2.43 SD higher (1.62 higher to 3.24 higher)
	240 (1 RCT)	ATQ	GVL	Difficult	120	120	-	-	SMD 0.71 SD lower (0.97 lower to 0.45 lower)
	327 (2 RCTs)	ATQ	MG	All	164	163	-	-	SMD 0.67 SD lower (1.2 lower to 0.14 lower)
	87 (1 RCT)	ATQ	MG	Normal	44	43	-	-	SMD 0.97 SD lower (1.42 lower to 0.52 lower)
	240 (1 RCT)	ATQ	MG	Difficult	120	120	-	-	SMD 0.43 SD lower (0.68 lower to 0.17 lower)
	45 (1 RCT)	AWS	MCL	All	23	22	-	-	SMD 0.89 SD lower (1.5 lower to 0.27 lower)
	70 (1 RCT)	AWS	GVL	All	35	35	-	-	SMD 0.97 SD lower (1.47 lower to 0.48 lower)
	70 (1 RCT)	AWS	GVL	Difficult	35	35	-	-	SMD 0.97 SD lower (1.47 lower to 0.48 lower)
	78 (1 RCT)	CM	MCL	Normal	39	39	-	-	SMD 1.53 SD lower (2.04 lower to 1.02 lower)
	90 (1 RCT)	CM	MCL	Difficult	45	45	-	-	SMD 0.76 SD lower (1.19 lower to 0.33 lower)
	56 (1 RCT)	GVL	MG	Normal	28	28	-	-	SMD 0.81 SD lower (1.36 lower to 0.27 lower)
	300 (2 RCTs)	GVL	MG	Difficult	150	150	-	-	SMD 0.32 SD higher (0.1 higher to 0.55 higher)
	193 (3 RCTs)	MCL	MG	Difficult	96	97	-	-	SMD 0.92 SD higher (0.62 higher to 1.22 higher)
	43 (1 RCT)	GVL	KV	Normal	22	21	-	-	SMD 0.82 SD lower (1.44 lower to 0.19 lower)
	60 (1 RCT)	GVL	MCL	Difficult	30	30	-	-	SMD 0.6 SD lower (1.12 lower to 0.08 lower)
Glottic view	1400 (16 RCTs)	ATQ	MCL	All	492/697 (70.6%)	674/703 (95.9%)	OR 45.41 (2.29 to 902.16)	706 per 1000	285 more per 1000 (from 140 more to 294 more)
	1014 (10 RCTs)	AWS	MCL	All	441/524 (84.2%)	483/490 (98.6%)	OR 8.60 (1.01 to 73.79)	842 per 1000	137 more per 1000 (from 1 more to 156 more)
	711 (6 RCTs)	AWS	MCL	Difficult	295/355 (83.1%)	349/356 (98.0%)	OR 49.84 (3.97 to 626.44)	831 per 1000	165 more per 1000 (from 120 more to 169 more)
Safety	313 (5 RCTs)	ATQ	GVL	All	112/156 (71.8%)	127/157 (80.9%)	OR 7.92 (1.93 to 32.47)	718 per 1000	235 more per 1000 (from 113 more to 270 more)
	181 (3 RCTs)	ATQ	GVL	Normal	61/90 (67.8%)	78/91 (85.7%)	OR 2.78 (1.13 to 6.81)	678 per 1000	176 more per 1000 (from 26 more to 257 more)
	190 (3 RCTs)	ATQ	MCL	Normal	37/94 (39.4%)	55/96 (57.3%)	OR 3.13 (1.26 to 7.80)	394 per 1000	277 more per 1000 (from 56 more to 441 more)
	353 (3 RCTs)	CM	MCL	All	168/175 (96.0%)	162/178 (91.0%)	RR 7.49 (1.62 to 34.61)	960 per 1000	1000 more per 1000 (from 595 more to 1000 more)
	130 (1 RCT)	CMD	MCL	Normal	21/65 (32.3%)	65/65 (100.0%)	OR 271.14 (13.2 to 5568.91)	323 per 1000	669 more per 1000 (from 540 more to 677 more)
	60 (1 RCT)	GVL	MG	All	19/30 (63.3%)	23/30 (76.7%)	OR 0.07 (0.01 to 0.61)	633 per 1000	525 fewer per 1000 (from 616 fewer to 120 fewer)

RCT, randomized controlled trial; CI, Confidence interval; OR, Odds ratio; SMD, Standardised mean difference; N/A, Not available; VL, videolaryngoscope; MCL, Macintosh laryngoscope; ATQ; Airtaq, AWS, Airwayscope; CM, C-MAC; CMD, C-MAC d-blade; GVL, GlideScope; KV, King Vision; MG, McGrath.

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
