# Peer review of "Comparisons of Videolaryngoscopes for Intubation Undergoing General Anesthesia: Systematic Review and Network Meta-Analysis of Randomized Controlled Trials"

_jpm, 2022, doi:10.3390/jpm12030363_

Round 1
Reviewer 1 Report
The authors have composed a manuscript outlining comparison of video laryngoscopes with regard to their likelihood of intubation success in patients undergoing general anesthesia. The overall findings of the meta-analysis concluded with moderate certainty that intubation success was more likely with video laryngoscopes compared to standard Macintosh direct laryngoscope blades. The overall more successful video laryngoscopes included the CMAC D-Blade and King vision, with the primary outcome variable designated as success rate of first attempt at endotracheal intubation as confirmed with capnography. Evidence demonstrating a difference between direct comparison of the various video laryngoscopes showed lower level of certainty. This is a relevant publication to the field of anesthesia given that prior meta-analyses evaluating videolaryngoscopes did not find evidence of superiority of videolaryngoscope compared with direct laryngoscopy. The authors should be commended for this contribution to the practice of anesthesiology and airway management at other clinical sites including in critical care units.
Positive attributes of the manuscript include adherence to PRISMA guidelines, registration of the study with PROSPERO, and Quality assessment and level of certainty using the risk of bias tool and the GRADE tool of each study included. The fact that this was performed independently by each of the reviewers and is described in detail under quality assessment I am certain took quite a bit of time but contributes to the overall validity of the findings of the study. I also appreciated that the authors submitted details of their search strategy in supplemental material, although these search engines are presently coded in a way that rarely permits precise repetition of an existing search, given that the engine itself tailors its search to the prior interests/searches of the reviewing party.
Although the article would benefit from a review by an English-speaking editor skilled in scientific writing, the organization of the manuscript is quite sound. The authors have also paid careful attention to defining terms (e.g. SUCRA) and including relevant data points in the supplementary material. However, I would re-work the discussion first paragraph as it seems a bit confusing to the reader that you state the CMD is more successful in “all and difficult” airways but the Kingvision is more successful in normal airway. Also the authors state 4 VLs were listed as the top 3 VLs and it is unclear what is meant by “each airway circumstance.” Although by careful reading of the manuscript a reader can parse out what is meant by these statements, many non-academic clinicians will exclusively read the discussion session therefore the conclusions drawn there should be able to stand alone.
Questions for the authors:
The authors have neglected to state if their search strategy underwent peer review. Please state the parties that were involved in construction of the search strategy and any other parties affiliated with providing input or formal peer review.
The type of bias present in each manuscript is noted, which is appreciated. However, the discussion of study limitations could potentially be specified a bit further. Different intubation experience levels could potentially be assessed across the studies (resident level study or attending level study) and I would also be interested to know if there was a different outcome in subgroup analysis of the double lumen versus single lumen intubation experience. I understand the authors have stated that many of these variables were not reported accurately (time of anesthesia, hemodynamic changes, use of other medications) but I imagine most studies will have reported if the ETT was single or double lumen. I only perseverate on this data point because it could be that certain VLs are more successful with DLT than single, which would change the study conclusions and potentially impact practice patterns of our readers.
An explanation for the difference in findings in this meta-analysis and those that preceded it may be warranted. Although I do not recommend the authors overstate the ramifications or benefits of their findings, an interpretation of why your meta-analysis found a benefit (i.e. more widespread use and availability of VLs in training programs) seems the most logical underlying factor.
Author Response
To. Reviewer #1
Thank you for your detailed and rigorous comments. We considered again for all of your comments as follows. Sincerely.
Best regard.

Reviewer 2 Report
Introduction:
The opening sentence requires some editing.
There are many potential complications of tracheal intubation among which there are dental injuries. Authors focused on three “complications” it is arguable if a sore throat is a complication or rather quite expected consequences of tracheal intubation.
Authors might consider grouping complications like respiratory, hemodynamics or mechanical so it be described in a more organized way.
Authors might consider refence Anesthesiology December 2012, Vol. 117, 1223–1233.
https://doi.org/10.1097/ALN.0b013e31827537cb for predictors of difficult airways.
There are many acronyms for laryngoscope devices I would suggest to use full name of device in the manuscript as it would facilitate manuscript comprehension.
The manuscript would benefit with a figure that would demonstrate all of the devices in one picture.
Outcome measure:
How authors concluded that intubation device was responsible for sore throat?
Where there a data available for size of ETT used?
There is a data available suggesting that smaller size ETT reduce the incidence of sore throat
Some of the devices require quite wide interincisor distance for insertion would a dental damage betere reflect on device safety than sore throat ?
If Cook modification of CL was used why grade 2b was considered a good-glottic view ?
Figure 2 a very interesting approach to summarize the outcomes from intubation devices.
Could authors provide a higher resolution of the figure?
Again a lot of abbreviations make it difficult to read
Could authors reflect of the clinical implication of their findings was a difference in intubation times and views clinically relevant ?
The authors did an enormous work of including 96 RCTs but the manuscript requires some major edits to improve comprehension and scientific robustness.
Author Response
To. Reviewer #2
Thank you for your detailed and rigorous comments. We considered again for all of your comments as follows. Sincerely.
Best regard.

Reviewer 3 Report
Comments attached in the file.

Author Response
To. Reviewer #3
Thank you for your detailed and rigorous comments. We considered again for all of your comments as follows. Sincerely.
Best regard.

Round 2
Reviewer 1 Report
The authors have improved upon the manuscript by addressing my concerns.
I do feel one section requires further revision:
This systematic review and network meta-analysis demonstrated that the CMD was relatively successful compared with other VLs and MCL for the tracheal intubation undergoing general anesthesia in all airway circumstances. And, while the KV was more successful in normal airway circumstances, the CMD was more successful in difficult airway circumstances compared with other VLs and MCL. Other VLs such as MG, GVL, AWS, and CM were top three ranked VLs for intubation success in all, normal and difficult airway circumstances.
"Airway circumstances" is not a terminology I am familiar with - I believe "reassuring airway exam, anticipated difficult airway" are more standard. However, there is a difference between anticipated difficult airway, known difficult airway (patient was a difficult airway in prior intubation attempts) and all of this is salient to the review. Would recommend authors re-examine how many of these airways fall into either category (concerning airway exam and thus anticipated difficult airway, unanticipated difficult airway) and how these groupings were determined.
Reviewer 2 Report
The authors have addressed all the issues pointed out in the review. Authors have put significant effort to improve the manuscript. The paper might benefit from additional language review. I would suggest to accept this manuscript for the publication.
Reviewer 3 Report
File attached.
